# Radiators Adjustment in Multi-Family Residential Buildings—An Analysis Based on Data from Heat Meters

Karol Bandurski *, Andrzej Górka and Halina Koczyk

Faculty of Environmental Engineering and Energy, Institute of Environmental Engineering and Building Installations, Poznań University of Technology, Berdychowo 4, 60-965 Poznań, Poland; andrzej.gorka@put.poznan.pl (A.G.); halina.koczyk@put.poznan.pl (H.K.)
* Correspondence: karol.bandurski@put.poznan.pl

**Abstract:** Energy is consumed in buildings through the use of various types of energy systems, which are controlled by the occupants via provided interfaces. The quality of this control should be verified to improve the efficiency of the systems and for the comfort of the occupants. In the case of residential buildings, due to privacy reasons, it is problematic to directly monitor human–building interactions using sensors installed in dwellings. However, data from increasingly common smart meters are easily available. In this paper, the potential use of data from heat meters is explored for the analysis of occupant interactions with space-heating (SH) systems. A pilot study is conducted based on a one-year set of daily data from 101 dwellings. First, the identification of an indoor temperature and a strategy for thermostatic radiator valve (TRV) adjustments for all the investigated dwellings is presented. Second, the performed analysis suggests that 96% of the households did not use the automatic adjustment function of the TRVs since adjustments using the on–off mode were the most common, which could be empirical evidence for Kempton's theory on mental models of home heating controls. The reasons for this could be the weakness of the TRV as an SH interface and the technical specificity of the analyzed SH (its supply temperature). The preliminary investigation confirms the potential of the proposed methodology, but further research is needed.

**Keywords:** heating; smart meters; TRV; occupant behavior; residential building thermostats

## 1. Introduction

Decreasing the energy used in residential buildings while maintaining high-level comfort is one of the major challenges that face the ongoing energy conservation transition, as declared by the European Union [1]. In working towards this goal, it is important to understand where and why energy is consumed by the users in the buildings [2]. Occupants use energy to achieve comfort, and they can continue to do so by taking advantage of HVAC (heating, ventilation, and air conditioning) with the provided building system interface. Research on how to achieve comfort [3], as well as the interactions with the system interface [4,5], is important because the integration of these two processes is crucial for the performance of the building. If the interface is not adapted to the requirements of the occupants, then achieving comfort will be difficult for them. The consequences of this would be a loss of trust in the control system [6], unexpected use of the building's systems, and/or unnecessary use of energy. Examples of these processes have been collected by O'Brien and Gunay [7] or Sarran et al. [8].

Basic residential building interfaces include windows, movable shades, thermostats, and light switches [5]. A simple Scopus search (date: 26 October 2023, search within: "Article title, Abstract, Keywords: "occupant behaviour" AND "model" AND "windows"; "occupant behaviour" AND "model" AND "blinds"; "occupant behaviour" AND "model" AND "thermostats"; "occupant behaviour" AND "model" AND "lighting") shows that the most common research relates to the control of windows (254 papers) and lighting

(152 papers), and much less frequently thermostats (56 papers) and shading (40 papers), which indicates that there is still a lack of recognition of these processes. It might seem that thermostat control, especially in terms of heating, can be simplified to controlling the temperature that is maintained in the building, which affects transmission heat loss. However, the way in which a water heating system is controlled significantly affects, for example, the distribution of heat loss in the system, which may increase the indoor temperature in an uncontrolled way and, in turn, may cause occupants to open a window. There is, thus, a relationship between the building's systems and the building envelope. In addition, the control of the heating system also influences the peak loads or the operation of the auxiliary equipment of the system. Therefore, an analysis of the use of heating systems is of great importance for future designs that reduce energy loss and peak loads.

A general overview of factors that influence the ways in which space heating (SH) is adjusted was prepared by Wei et al. [9]; such heating systems can be classified as air- or water-based. In general, the former heating type is more common in North America, and the latter is more common in Europe. Both forms can be controlled by smart thermostats, the usage of which is most described in the literature for SH interfaces due to remote data collections performed by their manufacturers [10–15]. However, in, for example, Poland, the Czech Republic, and Bulgaria [16], but also Germany and the UK, the most common interfaces for SH are thermostatic radiator valves (TRVs); to illustrate this point, it is known that 45% of households in Poland have a TRV [17]. Lomas et al. [18], in their critical review, explained that the energy-saving effect of thermostats on energy consumption for SH in residential buildings is not well documented. Their studies demonstrate that the energy efficiency of the zonal temperature control is moderately documented. They consider the depth of research on the savings that result from TRV installations and the superiority of smart thermostats over standard thermostats to be of a low rate. In a subsequent paper, Lomas et al. confirmed the contribution of TRVs towards a small amount of energy savings because of SH zonal control [19]; this study used the latest generation of TRVs, which contain electronics with remote control functionality. Another point of view in the evaluation of TRVs is provided by Cholewa et al. [20]. They described six years of heat consumption measurements for the SH requirements of, among others, three multi-family buildings, where conventional radiator valves were replaced with TRVs. Their measurements refer to the three years of energy consumption before and the three years after the valve replacement. The average saved energy, due to their use and the system's hydraulic balancing, was 17%.

To conclude the above findings, there is still a relatively low amount of research focusing on human interactions with heating controls. One of the most popular heating controls is the TRV, the energy efficiency of which is questionable. Therefore, there is a need to better understand its usage by occupants and to assess its usability for energy-efficient building operations.

In regard to the TRV–occupant interactions, the literature is limited to surveys [21–23] or measurement research based on only a selection of households [19,24,25], which involve a lot of additional sensors inside the dwellings. The authors have found only one expectation: a monitoring campaign conducted in the UK, where 47 flats were observed using smart heating control systems, which were implemented in buildings based on funding support from a European project [25]; however, the presented results refer to a specific group of users: elderly residents in a care home. From the point of view of continuous control of the energy efficiency of the operation of heating systems, these are not suitable methods of monitoring. The first method is subjective [26] and is limited to only collecting information once, as later repetition may be tedious for respondents. The second method requires an advanced monitoring system installed in the dwellings. The collected data are very informative but could be seen as personal data, which would not be acceptable for most residents. In the case of smart heating controls, acquiring the funds for such upgrades is problematic. Therefore, a methodology for analyzing the operational quality of the heating systems with TRVs or others based on water heaters that involve readily available data, such

as heat meters, would be valuable. Such approaches will be objective and will not disturb household privacy. Moreover, the collected data will not contain any direct information about the indoor environment of the dwellings. To date, methods for heat meter data analysis have been developed solely for studying the overall energy performance of a building [27–29].

The questions for this research are: (1) What information regarding the occupant control of the SH can be extracted from heat meters? and (2) Is it possible, based on this information, to learn more about people's use of TRV? Do they use TRV in accordance with its design? This paper proposes a methodology for determining parameters characterizing occupant–SH system interactions based on radiator model fitting to heat meter data. Therefore, the privacy zone of the occupants is not disturbed. Preliminary results are presented using data from a multi-family housing estate. Additionally, the methodology includes an algorithm for separating the heat consumption of the SH and the domestic hot water (DHW) from the dwelling heat meter data. Correlation between the dwelling characteristics and the observed TRV usage models is investigated.

## 2. Methodology

### 2.1. Research Object

Data for the analysis was collected from a multi-family housing estate in Poland, in a humid continental climate with mild summers and rainfall all year round, Dfb according to Köppen-Geiger climate classification [30]. The building estate was constructed at the beginning of the 21st century (Figure 1). It consists of 4 buildings with 108 apartments with a total area of 6752 m². The six-story buildings, the first story of which is a basement and the final two are two-story apartments, are represented in Figure 2. Table 1 summarizes the parameters of the dwelling types.

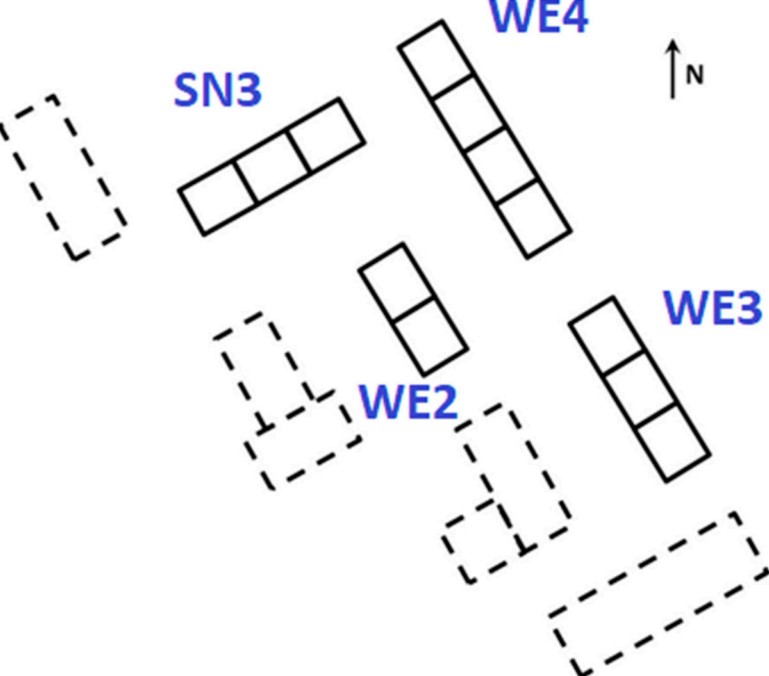

**Figure 1.** Plan of the studied housing estate. Solid lines represent the surveyed buildings and their staircases, and the dashed lines correspond to the buildings surrounding the housing estate. Adapted with permission from ref. [31].

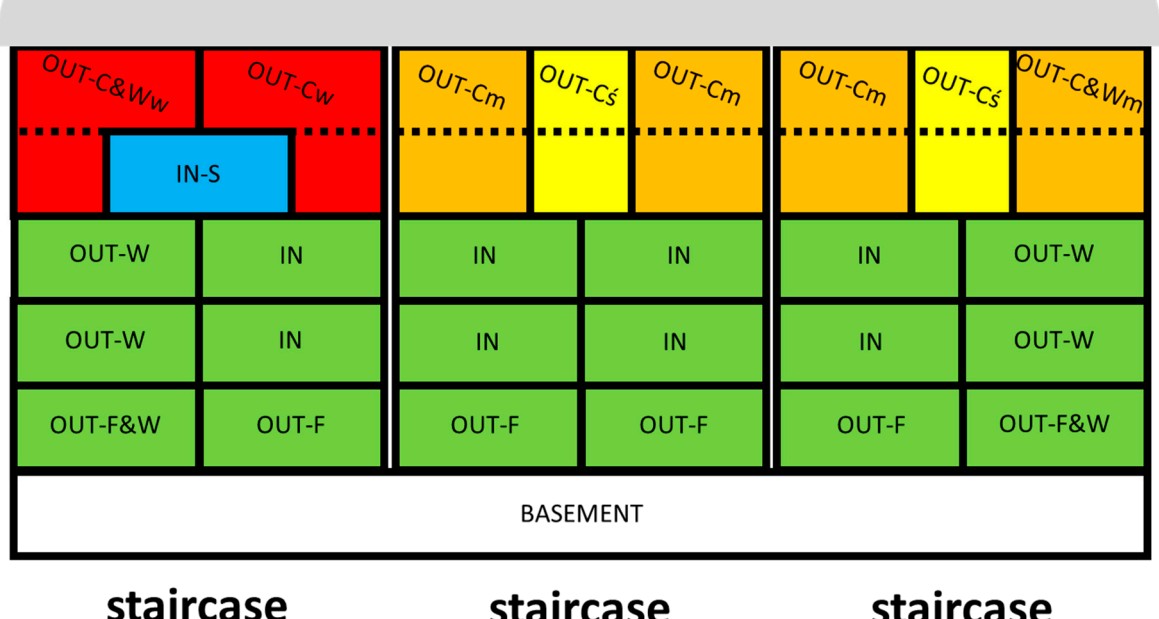

**Figure 2.** Types of dwellings in the analyzed buildings. Colors represent dwellings with identical floor areas. Two-story apartments are located on the top floor (except IN-S). The figure shows the layout of the three staircase buildings, for which the two-staircase building has an A and a B staircase, and the four-staircase building has an A staircase and three B staircases. Adapted with permission from ref. [31].

**Table 1.** Characteristics of the surveyed dwellings (see Figure 2 for the type of dwelling notation). Adapted with permission from ref. [31].

| Types of Dwellings | $A_f\,[\mathrm{m^2}]$ | $H_{tr}\,[\mathrm{W/K}]$ | $C_m\,[\mathrm{kJ/K}]$ | $A_{win}\,[\mathrm{m^2}]$ | $\dot{Q}_{D,SH,max}\,[\mathrm{W}]$ |
|:---:|:---:|:---:|:---:|:---:|:---:|
| IN | 53 | 22.54 | 29.98 | 9.04 | 1180 |
| IN-S | 36 | 17.27 | 16.65 | 6.78 | 880 |
| OUT-W | 53 | 31.81 | 29.88 | 9.87 | 1510 |
| OUT-Cm | 83 | 38.56 | 46.44 | 13.34 | 2410 |
| OUT-Cs | 79 | 42.20 | 35.88 | 15.84 | 2250 |
| OUT-Cw | 104 | 49.75 | 55.58 | 16.73 | 2990 |
| OUT-F | 53 | 30.76 | 29.98 | 9.04 | 1570 |
| OUT-CWm | 83 | 56.96 | 46.70 | 15.54 | 3060 |
| OUT-CWw | 104 | 68.30 | 55.37 | 18.93 | 3630 |
| OUT-FW | 53 | 40.03 | 29.88 | 9.87 | 1920 |

*2.2. Residential Thermal Stations System, Heat Network, and Gas Boilers*

Heating for the housing estate is provided by the local gas boilers located in the WE3 building. An underground heating network (HN) supplies the heated water from the gas boilers to WE2, WE4, and SN3 buildings. The heating network supplies the water temperature ($t_{\mathrm{HN},sup}$) to the residential thermal stations (RTSs), which all year round equals approximately 70 °C. The RTS are single-function residential thermal stations in which the DHW is prepared or the water is passed through to supply the SH. This solution reduces the distribution heat losses due to the same pipes being used for the SH and DHW distributions.

However, it enforces the maintained high supply temperature all year round to enable the preparation of the DHW.

### 2.3. Data

Measurements were performed from March 2015 to February 2016. Figure 3 shows a scheme depicting the research facility with the variables measured, estimated, and determined through a data analysis described in this paper.

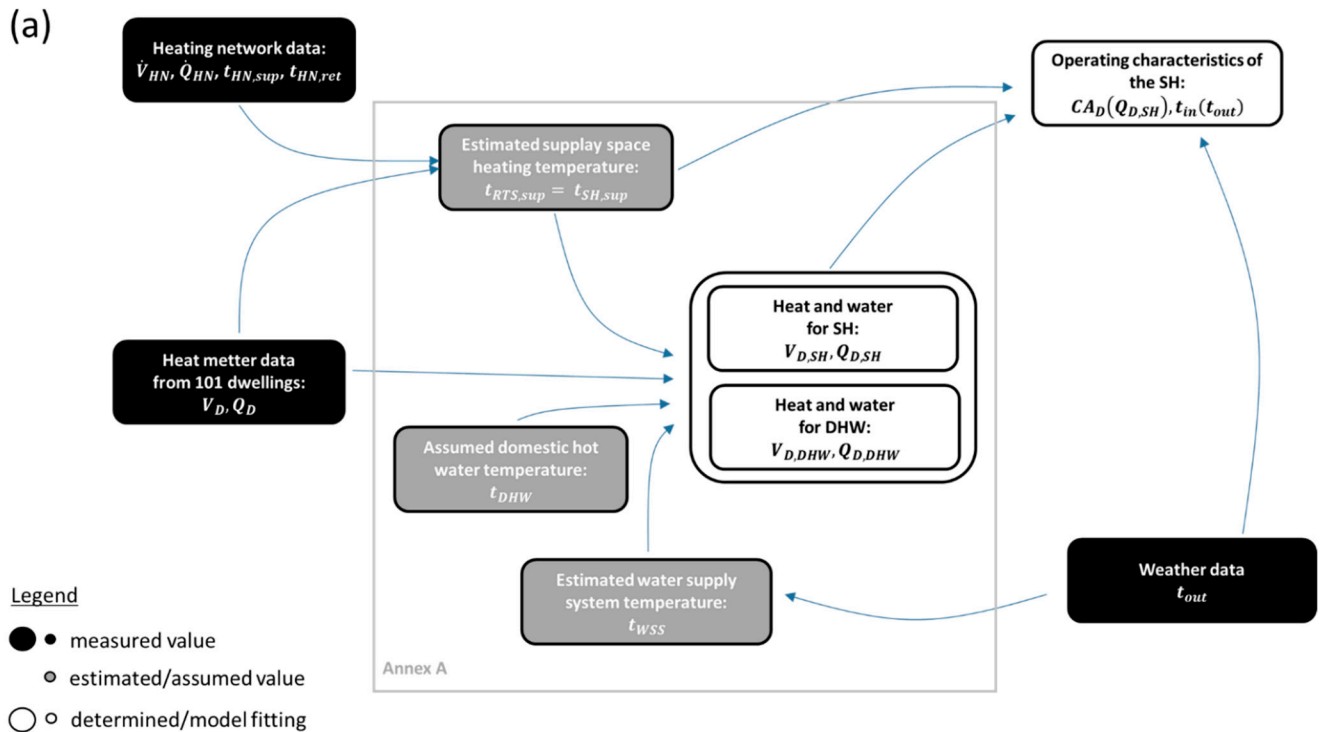

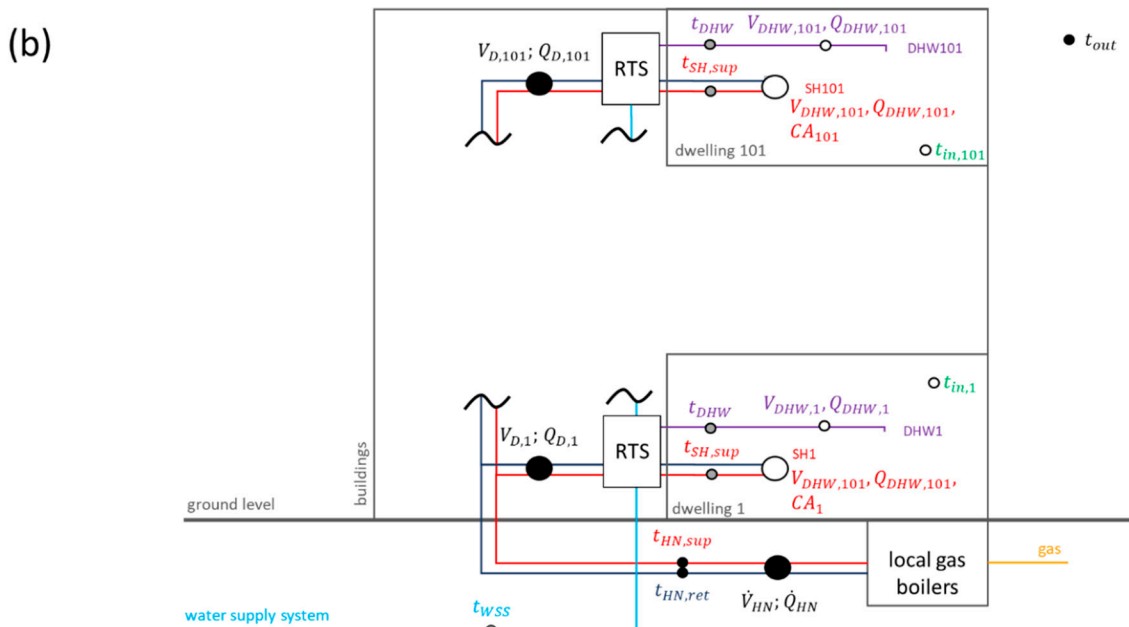

**Figure 3.** Scheme of (**a**) the performed analysis and (**b**) the research facility. Indication of the variables measured, estimated, and determined by a model fitting.

### 2.3.1. Heat Consumption and Heating Water Flow

Data from the dwellings was collected via Kamstrup MULTICAL302 heat meters, which were installed in 101 apartments. The remaining four apartments were not included in the analysis. The heat meter memory contained 460 daily records of the heat consumption and the heating water consumption of the dwelling. The memory also contained records with hourly and monthly data, but they were not included in the analysis. It is noteworthy that this type of heat meter, which is widespread, has a standard memory resolution (0.01 GJ), which makes daily analysis difficult and hourly analysis impossible. The problem is noticeable in the results (Section 3): models for the dwelling with very low heat consumption represent a sharp decrease in accuracy. The low resolution arises from the limited internal memory of the meters. However, it can be modified during production (subject to customer orders). This dataset is part of the ASHRAE Global Occupant Behavior Database [32,33], which is an open database.

Data on the gas boiler operations was obtained from the operator (Veolia Energia Poznań SA, Poznań, Poland). The data are provided in 15-min timesteps and include the supplied and return water temperature, the instantaneous thermal power of the boilers, the instantaneous heating water flow, the total heat transferred to the heat network, and the total volume of the heating water that flowed through the HN.

### 2.3.2. Outdoor Air Temperature

Outdoor air temperatures for the analysis were measured at a weather station located 3 km from the buildings under the study (Figure 4). Figure 5 compares outdoor temperature distribution in the city, taking measurements from 2015 onwards. It can be seen that, in 2015 and 2016, this does not differ significantly in terms of average, median, 25th, and 75th centile values from other years. The only noticeable difference is that one of the warmest winters was observed in 2015; however, this season is not included in the analysis (see Figure 4).

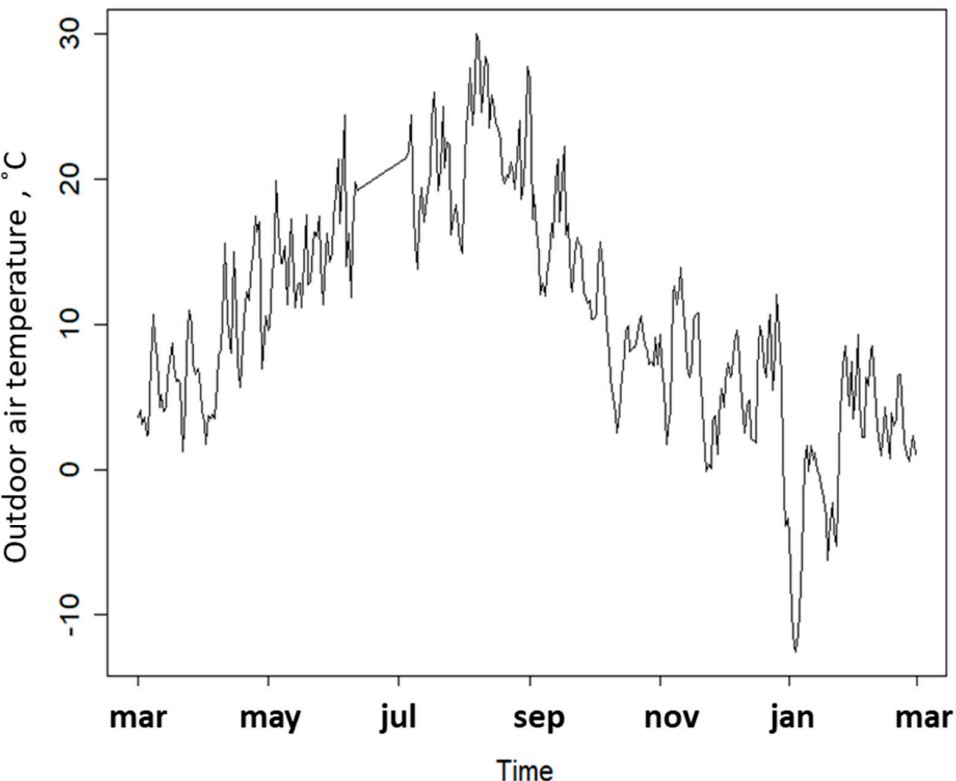

**Figure 4.** Outdoor air temperature. The end of June represents a period in which the measuring system failed. The straight lines provide an interpolation of the missing data.

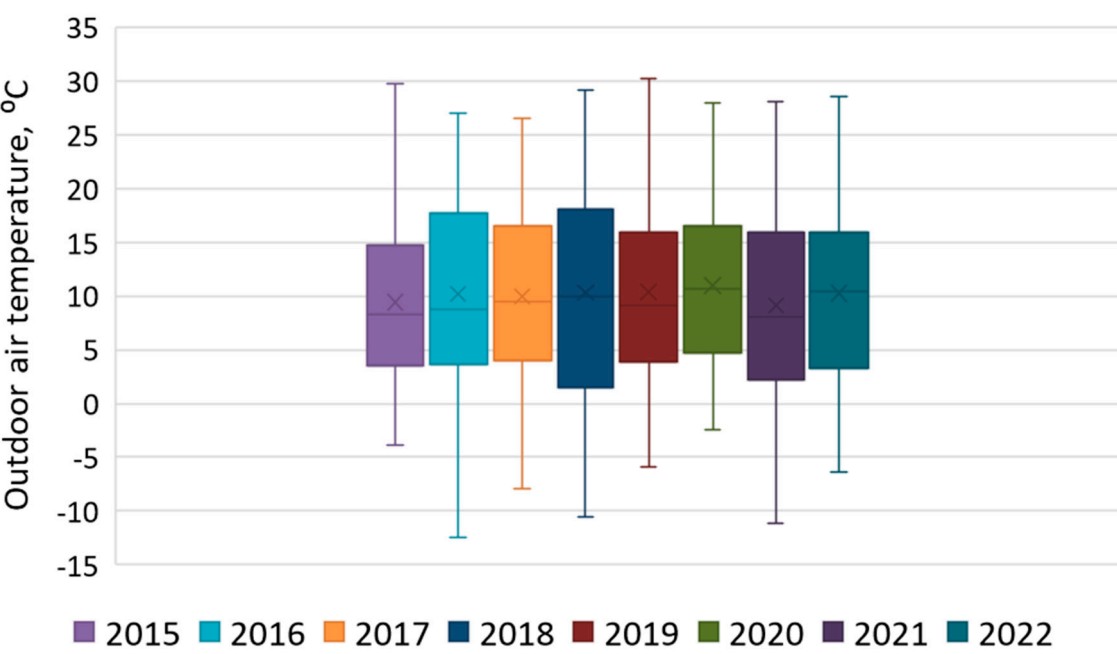

**Figure 5.** Yearly outdoor air temperature distribution, based on mean daily data, in the city where heat consumption data were collected.

### 2.4. SH Usage Model

For the analysis, only SH data were used. The methodology of the separation of heat and heating water consumption for the SH and the DHW from heat meter readings is described in Supplementary Material S1.

The power of the radiator, but also that of the SH, can be described by the following model [34–37]:

$$\dot{Q}_{D,SH} = \dot{m}_{D,SH} \cdot c_w \cdot \left(t_{SH,sup} - t_{in}\right) \cdot \left\{ 1 - \left[ \frac{\dot{m}_{D,SH} \cdot c_w}{\dot{m}_{D,SH} \cdot c_w + n \cdot CA_D \cdot \left(t_{SH,sup} - t_{in}\right)^n} \right]^{1/n} \right\} \quad (1)$$

Based on this research, it is possible to determine the values of the following variables found in Equation (1):

- $\dot{m}_{D,SH}$, the mean daily mass flow rate of the heating water in a given dwelling that is determined on the basis of the following expression:

$$\dot{m}_{D,SH} = \frac{V_{D,SH}}{24 \cdot 3600} \rho_w \quad (2)$$

in which the daily heating water consumption ($V_{D,SH}$) was measured by a heat meter, and the water densities ($\rho_w$) were calculated according to ref. [38] for the mean water temperature in the HN $\left(t_{HN,sup} + t_{HN,ret}\right)/2$,

- $c_w$, the water-specific heat capacity was calculated according to ref. [38] for the mean water temperature in the HN,
- $t_{SH,sup}$, the mean daily supply water temperature of SH; its determination is described in Supplementary Material S1, Equation (S1.1),
- $n$, the radiator type exponent is assumed to be 0.3, which is the same value for all radiators based on the analysis of the technical documentation and the information from the manufacturer.

The values of $t_{in}$ and $CA_D$ are unknown. The next section proposes the possible methods for controlling the heating system; a set of corresponding assumptions for these values are included.

### 2.5. Assumption for Occupant–SH Interaction

Two variables in Equation (1) determine how the SH is adjusted. These are:

- $t_{in}$, the temperature in the dwellings,
- $CA_D$, the product of the radiator heating area ($A_D$) in the dwelling and the coefficient of the heat transfer intensity between the radiators and the indoor environment.

The following proposed values for these variables are based on occupant behavior assumptions, which will later be verified based on the collected data.

#### 2.5.1. Indoor Temperature in the Dwelling $t_{in}$

The temperature in the dwellings was not measured, but data can be found in the literature regarding the temperature maintained in dwellings as a function of the outdoor temperature [39–46]. The occupant behavior assumptions for this analysis were prepared based on the data collected in multi-family buildings in Estonia [47], whose building cultural and climatic context is similar to that of Poland. Five models for the indoor temperature adjustment as a function of the outdoor temperature are proposed (Figure 6):

$$t_{var,high} = \max(-0.05t_{out} + 24.57, 0.33t_{out} + 20.17) \tag{3a}$$

$$t_{con,high} = \max(24, 0.33t_{out} + 20.17) \tag{3b}$$

$$t_{con,medium} = \max(22, 0.33t_{out} + 18.67) \tag{3c}$$

$$t_{con,low} = \max(20, 0.33t_{out} + 17.17) \tag{3d}$$

$$t_{var,low} = \max(0.05t_{out} + 19.57, 0.33t_{out} + 17.17) \tag{3e}$$

Models with a *var* subscript assume a change in indoor temperature during the heating season, while models with a *con* subscript assume that (during the heating season) indoor temperature is relatively constant.

#### 2.5.2. Radiators Operation Adjustment $CA_D$

The way that the TRV operates provides:

- hydraulic balancing of the system by the installers,
- opening and closing of the heating water flow through radiators via manual adjustment by the occupants,
- automated maintenance of the thermal comfort conditions in a room set by the occupants.

The final function is the basic energy-saving feature of the TRV. It operates thanks to an internal mechanism that smoothly regulates the heating water flow through the radiator as a function of the room temperature. In the most common variant, the TRV mechanism is based on the thermal expansion phenomenon of the substance; it does not require an external power supply. For a given setting of the TRV, the power of the radiator is adjusted in a manner that maintains the set temperature of the room. Usually, the TRV does not have precise temperature value settings, but only those over the range from 1 to 5, since the resulting internal temperature that the TRV maintains also depends on the location of the heater in the room.

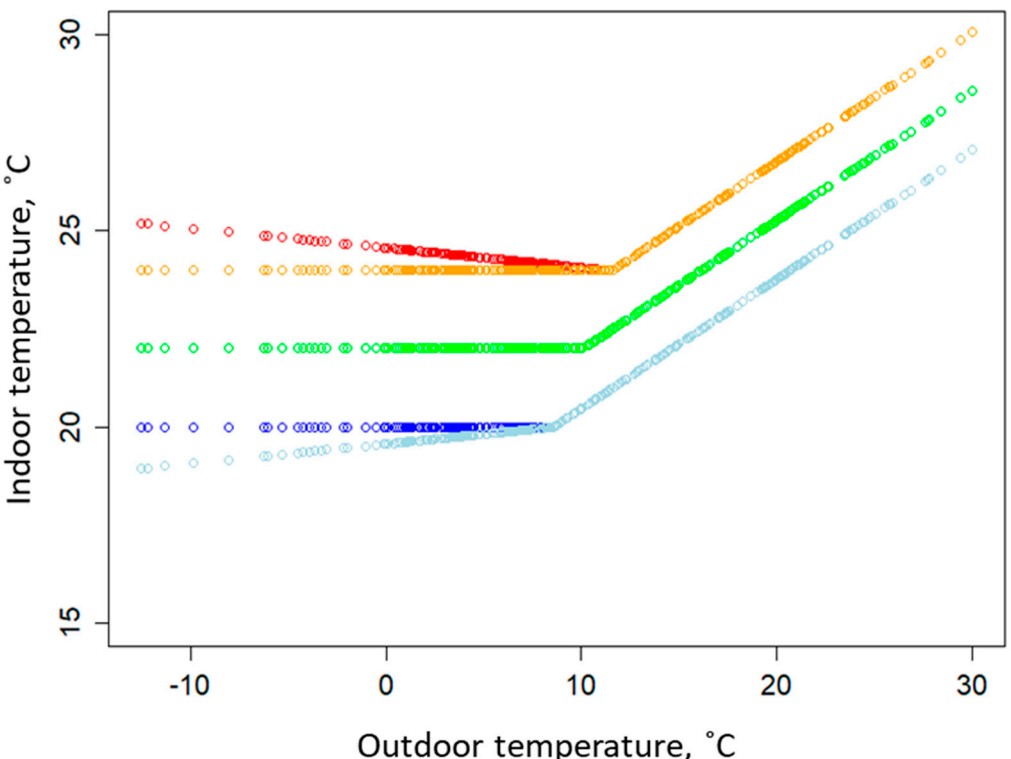

**Figure 6.** Analyzed models for the indoor temperature adjustment as a function of outdoor temperature: $t_{var,high}$ (red, Equation (3a)), $t_{con,high}$ (orange, Equation (3b)), $t_{con,medium}$ (green, Equation (3c)), $t_{con,low}$ (blue, Equation (3d)), and $t_{var,low}$ (light blue, Equation (3e)).

Each dwelling is equipped with radiators with known maximum power and type; also, the design temperature difference between the radiator and room is known. The following expression was used to calculate the maximum product of the radiator heating area and the coefficient of the heat transfer intensity ($CA_{D,max}$) base on these data:

$$CA_{D,max} = \frac{\dot{Q}_{D,SH,max}}{\Delta t_{SH-in,log}^{1+n}} \qquad (4)$$

There are two opposite ways to control the heating system:

- Automated, in which the radiator/SH power control is based on the automated control of flow through the TRV. Hence, the heating area of the radiators and the number of active radiators are not changed. Only power output from the radiator is regulated by its temperature. In this approach, the heating area of the system is fixed so that:

$$CA_{D,con} = const \qquad (5)$$

- Manual, in which the power control of the radiators/SH is based on manually opening and closing the given radiators. This is performed without the use of the TRV function that adjusts the flow through the radiator to the set temperature and the actual temperature in the room. Then, the control of the radiator/SH power is based on an adjustment of the heating surface of the radiators, which can be simplified as:

$$CA_{D,var} = CA_{D,max} \frac{Q_{D,SH}}{max(Q_{D,SH})} \qquad (6)$$

Equations (5) and (6) define the two proposed intensity adjustment models for the SH operation. Additionally, the heating system can be only partially used in a given dwelling, even with the smallest loads, in which case the maximum heating area of the

radiators is smaller than the available one ($A_{D,max}$). It is also possible that the heat transfer coefficient between the radiator and the surroundings, $C$, will be more intense compared to standard operations due to the simultaneous ventilation (by opening windows) and space heating. On this basis, the following assumptions for regulating the intensity of the radiator operation have been proposed:

- Automated:

$$CA_{D,con,v.high} = 1.50\, CA_{D,max} \tag{7a}$$

$$CA_{D,con,high} = 1.25\, CA_{D,max} \tag{7b}$$

$$CA_{D,con,medium} = CA_{D,max} \tag{7c}$$

$$CA_{D,con,low} = 0.75\, CA_{D,max} \tag{7d}$$

$$CA_{D,con,v.low} = 0.5\, CA_{D,max} \tag{7e}$$

- Manual:

$$CA_{D,var,v.high} = 1.50\, CA_{D,var} \tag{7f}$$

$$CA_{D,var,high} = 1.25\, CA_{D,var} \tag{7g}$$

$$CA_{D,var,medium} = CA_{D,var} \tag{7h}$$

$$CA_{D,var,low} = 0.75\, CA_{D,var} \tag{7i}$$

$$CA_{D,var,v.low} = 0.5\, CA_{D,var} \tag{7j}$$

All the multipliers (from 0.5 to 1.5) refer to the change in the heating area ($A_D$) or change in the coefficient of the intensity of heat exchange between the radiator and the surrounding environment ($C_D$). The heating area should not be higher than the available heating area in the dwelling. Therefore, multipliers above 1 suggest an intensification of $C_D$, e.g., by intense ventilation of the heated spaces.

*2.6. Method of Estimation of the Occupant–SH Interaction Characteristic*

To assess which of the control methods is most likely to be applied in a given dwelling, calculations based on Equation (1) and all combinations of the above-described assumptions are compared with the measured heat consumption given from the following:

$$\dot{Q}_{D,SH,meas} = \frac{Q_{D,SH}}{24 \times 3600} \tag{8}$$

Here, the root mean square errors (RMSEs) are used as indicators that assess the fitting models with a given occupant–SH interaction assumption, i.e.,:

$$RMSE = \frac{\sqrt{\left(\dot{Q}_{D,SH,meas} - \dot{Q}_{D,SH}\right)^2}}{\dot{Q}_{D,SH,meas}} \tag{9}$$

## 3. Results

### 3.1. Model for All the Buildings

First, a model for the whole housing estate is investigated by summing the heat and heating water consumption from all the metered dwellings. The best-fit models are based on assumptions $CA_{D,var,medium} + t_{con,medium}$, which represent the middle of the considered values. The RMSE of the model predictions is 7% (Figure 7). The following models have a similar accuracy: $CA_{D,var,medium} + t_{con,high}$ (RMSE = 8%) and $CA_{D,var,medium} + t_{var,high}$ (RMSE = 9%).

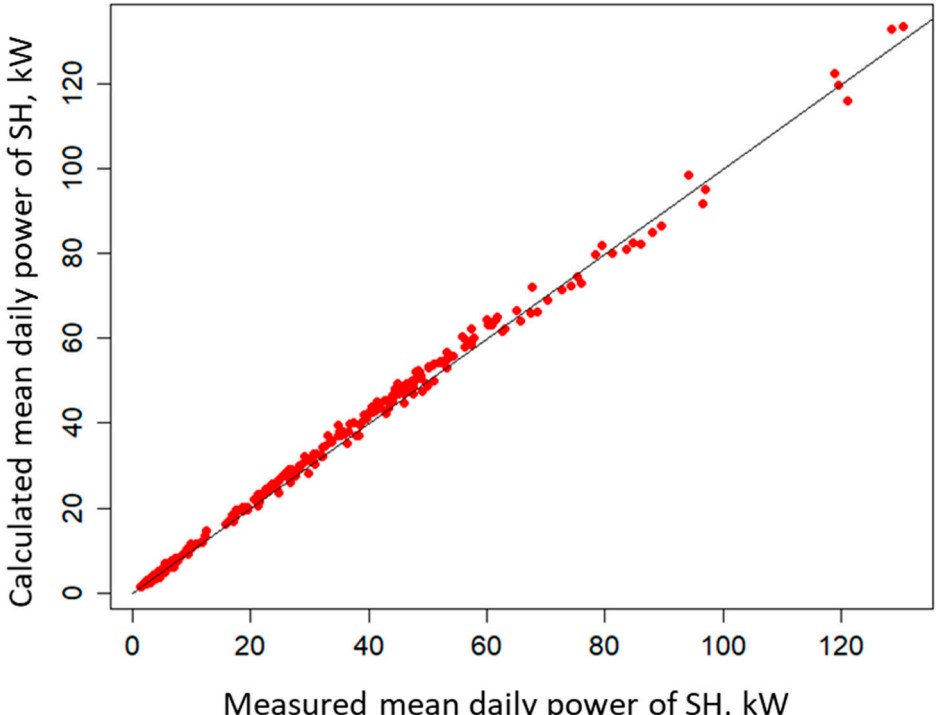

**Figure 7.** Measured values of the mean daily heat power delivered to the metered apartments of the housing estate compared with those predicted by the best-fit model ($CA_{D,var,medium} + t_{con,medium}$).

### 3.2. Dwellings Models

Table 2 shows several models of a given type found for the 101 investigated dwellings. The graphical comparison of the model results with the measurements for each dwelling is presented in Supplementary Material S2. The heating system control in the largest number of dwellings is described by the models $CA_{D,var,medium} + t_{con,medium}$ and $CA_{D,var,medium} + t_{con,high}$; the 17 dwellings are fitted for each.

The fitting of the models is shown as a histogram in Figure 8. Here, it can be seen that more than half of the models have an RMSE of less than 30%, and 80% of the models have an RMSE of less than 50%. We take RMSE < 25% as the range of the most reliable models (the red line in Figure 8). The accuracy of the models is correlated exponentially with the dwelling's heat consumption for the SH (Figure 9). This means that, for households with low heat consumption, low heat meter resolution causes a high RMSE (see Section 2.3.1).

Among the occupant–SH interaction models investigated, only 4% of the dwellings were fitted with models with a control based on automated flow adjustment by the TRV, and none of these have RMSE values lower than 25%. The models of the remaining dwellings suggest that the SH operation control is mainly based on the control of the operation time and the number of radiators (models $CA_{D,var,...}$): 69% (81% for RMSE < 25%) of the dwellings controlled the radiator area over the whole range of design heating areas ($CA_{D,var,medium}$), and in 28% (17% for RMSE < 25%) of the dwellings $CA_D$ is larger than the designed one. As noted in Section 2.5.2, this could be due to a larger coefficient $C$, which

represents a more intense heat transfer between the radiator and the indoor environment compared with standard conditions, e.g., those caused by simultaneous ventilation and heating of the dwellings.

**Table 2.** Number of dwellings for which a model with given assumptions is fitted. The value in brackets is the number of models for which RMSE < 25% is provided.

|  | $t_{var,high}$ | $t_{con,high}$ | $t_{con,medium}$ | $t_{con,low}$ | $t_{var,low}$ | Total |
|---|---|---|---|---|---|---|
| $CA_{D,var,v.high}$ | 0 | 0 | 0 | 3(0) | 0 | 3(0) |
| $CA_{D,var,high}$ | 2(0) | 2(2) | 8(1) | 4(2) | 8(3) | 24(8) |
| $CA_{D,var,medium}$ | 11(7) | 17(12) | 17(11) | 11(4) | 13(5) | 69(39) |
| $CA_{D,var,low}$ | 0 | 0 | 0 | 0 | 1(1) | 1(1) |
| $CA_{D,var,v.low}$ | 0 | 0 | 0 | 0 | 0 | 0 |
| $CA_{D,con,v.high}$ | 0 | 1(0) | 0 | 0 | 0 | 1(0) |
| $CA_{D,con,high}$ | 0 | 0 | 0 | 0 | 0 | 0 |
| $CA_{D,con,medium}$ | 0 | 0 | 0 | 0 | 0 | 0 |
| $CA_{D,con,low}$ | 0 | 0 | 0 | 0 | 1(0) | 1(0) |
| $CA_{D,con,v.low}$ | 0 | 2(0) | 0 | 0 | 0 | 2(0) |
| Total | 13(7) | 22(14) | 25(12) | 18(6) | 23(9) | 101(48) |

Indoor temperature adjustment in a dwelling is definitely more varied. The most widely used is the mean internal temperature profile, $t_{con,medium}$, which has been adjusted for the whole housing estate, employed in 25% (25% for RMSE < 25%) of the dwellings. The most rarely used profile is the one in which, for the duration of the heating period, there is an increase in the internal temperature compared to the transitional season ($t_{var,high}$); this applies to 13% (14% for RMSE < 25%) of the apartments.

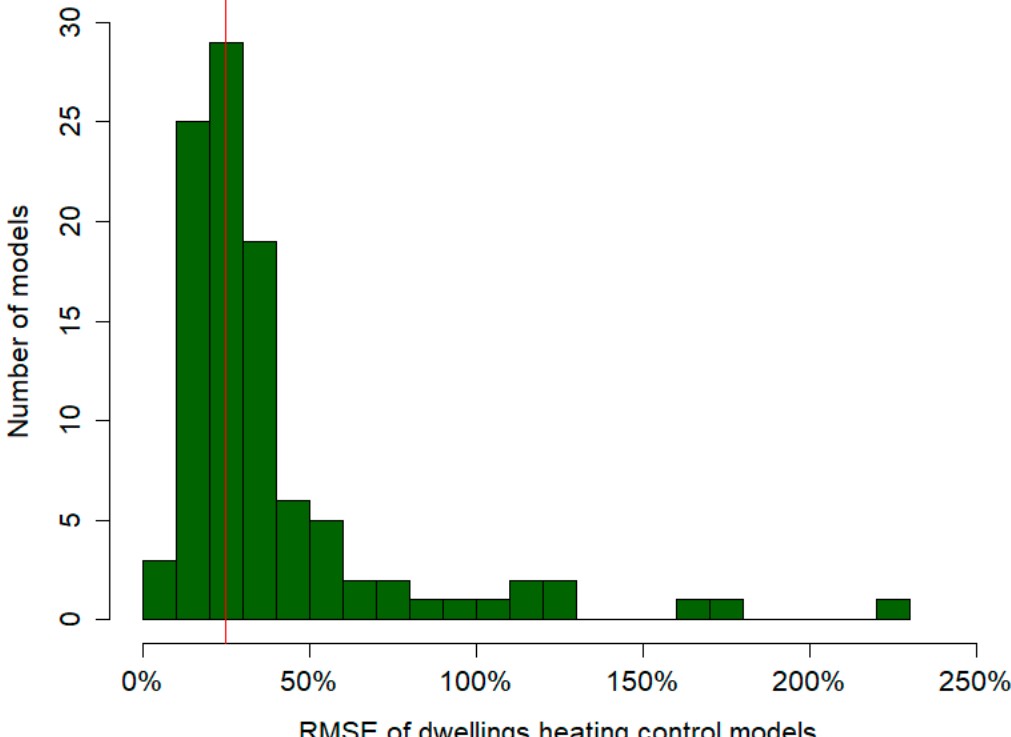

**Figure 8.** Histogram of the RMSE values for the dwelling models. The red line indicates the position of 25% RMSE, below which represents the most reliable models.



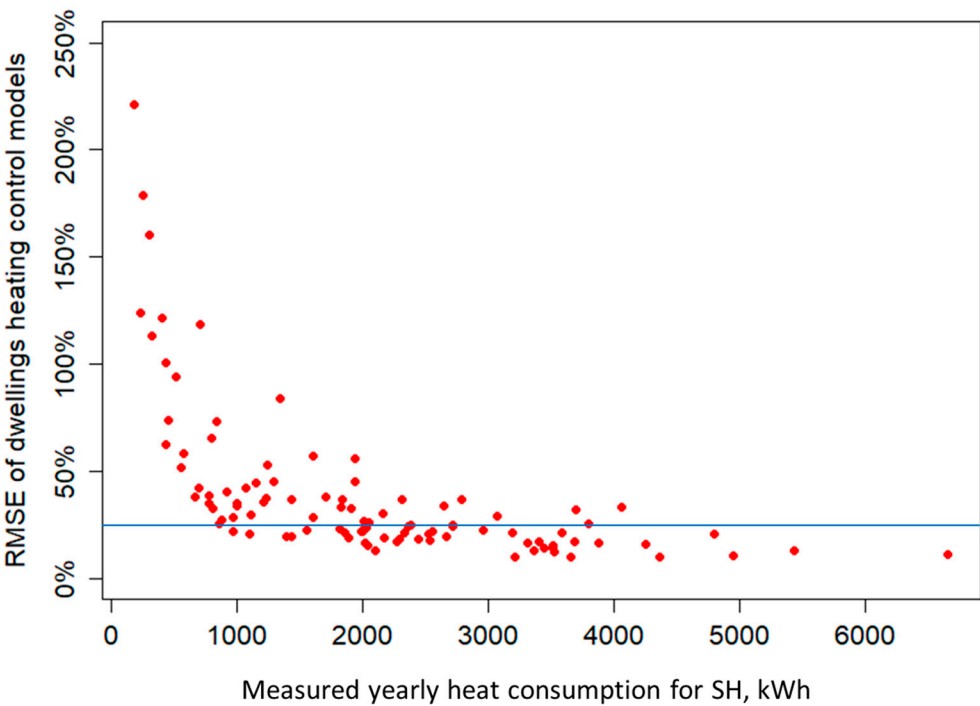

**Figure 9.** RMSE values of the dwelling models against the yearly heat consumption of the dwellings. The blue line indicates the position of 25% RMSE, below which represents the most reliable models.

### 3.3. Models of Occupant–SH Interaction and Dwelling Energy Characteristics

The $\chi^2$ coefficient is used to check if there is a correlation between the two fitting models and the floor on which the dwelling is located, the heat transmittance of the dwelling, and the block to which the dwelling belongs. Only one of the six cases is statistically confirmed to correlate, i.e., the *p*-value of the test is less than 0.1. The correlation between the model of the internal temperature and the block where the apartment is located has the value $p = 0.06$. Table 3 summarizes the number of indoor temperature models in each block. It can be seen here that only within building WE3 are the proportion of the models with higher internal temperatures ($t_{var,high}$ and $t_{con,high}$) greater than those with lower ones ($t_{var,low}$ and $t_{con,low}$). In this building, there is a gas boiler room in the basement, so, on one hand, it is an object in which the heat losses from the boiler room heat a part of dwellings and, on the other hand, $t_{SH,sup}$ is higher than in the other buildings. From the results presented, it seems more important that the heat losses from the boiler room cause higher temperatures in some dwellings. Introducing a significantly lower $t_{SH,sup}$ in the model for those dwellings causes the temperature in the apartments to be lower than in reality. In other words, the effect of specifying an underestimated $t_{SH,sup}$ in the model of these dwellings would produce an overrepresentation of the dwellings with low indoor temperatures rather than higher temperatures.

**Table 3.** Number of dwellings for which the indoor temperature model is fitted by building type. The boiler is located in WE3, so this is where the highest supply temperature of the RTS resides.

|  | $t_{var,high}$ | $t_{con,high}$ | $t_{con,medium}$ | $t_{con,low}$ | $t_{var,low}$ | **Total** |
|---|---|---|---|---|---|---|
| WE2 | 1 | 2 | 5 | 4 | 5 | 17 |
| WE3 | 5 | 8 | 4 | 7 | 1 | 25 |
| SN3 | 6 | 3 | 5 | 3 | 8 | 25 |
| WE4 | 1 | 9 | 11 | 4 | 9 | 34 |

### 3.4. Models of Occupant–SH Interaction and Heat Consumption

The method by which the SH is adjusted may influence the heat consumption of the dwelling. However, the coefficient of the heat transmittance of the dwelling and, in the case of multi-family buildings, the heat exchange between the adjacent dwellings are also meaningful. Figure 10 shows the annual heat consumption of the dwellings described by each assumption. Looking at the median and mean values of the SH operation intensity, the models with higher values have a lower heat consumption per floor area, but these do not have statistically significant differences according to the ANOVA test, i.e., $p > 0.1$. In the case of the indoor temperature models, the effect on heat consumption is more pronounced; it is statistically significant according to the ANOVA test, i.e., $p < 0.1$. However, it is difficult to explain from the physical point of view because the highest heat consumption occurs for dwellings with mean indoor temperature values.

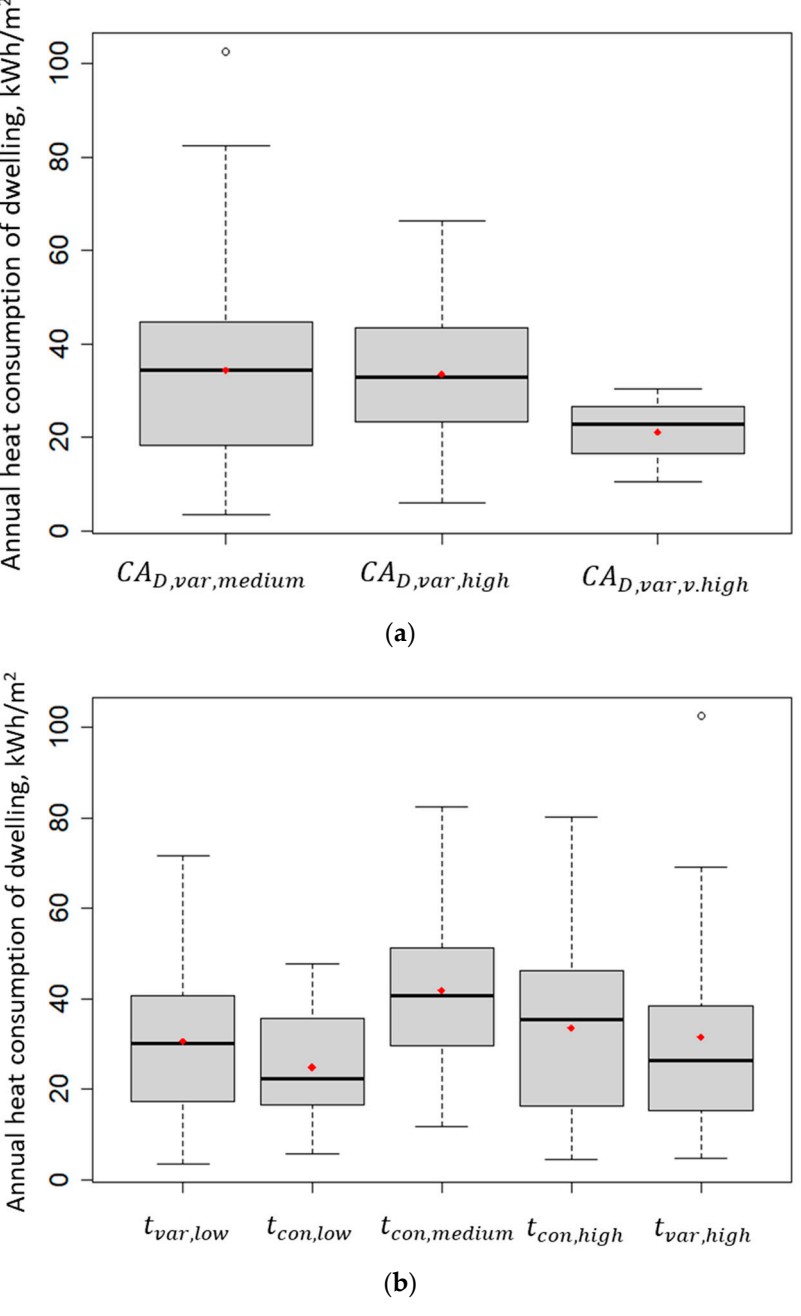

**Figure 10.** Annual heat consumption of the dwellings for SH purposes based on models with (**a**) SH operation intensity and (**b**) indoor temperatures.

Figures 11 and 12a show the length of the heating season for the models. Only the relationship between the indoor temperature model and the length of the heating season is statistically significant (ANOVA test, $p < 0.1$). On the one hand, the statistical significance vanishes (ANOVA test, $p > 0.1$) after limiting the set of models to the most reliable ones (RMSE < 25%). In contrast, examining only the models at $t_{con,...}$ and their medians (Figure 12b), the relationship is easier to interpret physically: the longer heating seasons are observed for the dwellings in which higher indoor temperatures are maintained. Next, the $t_{var,...}$ models imply a more specific use of the SH, which requires more research.

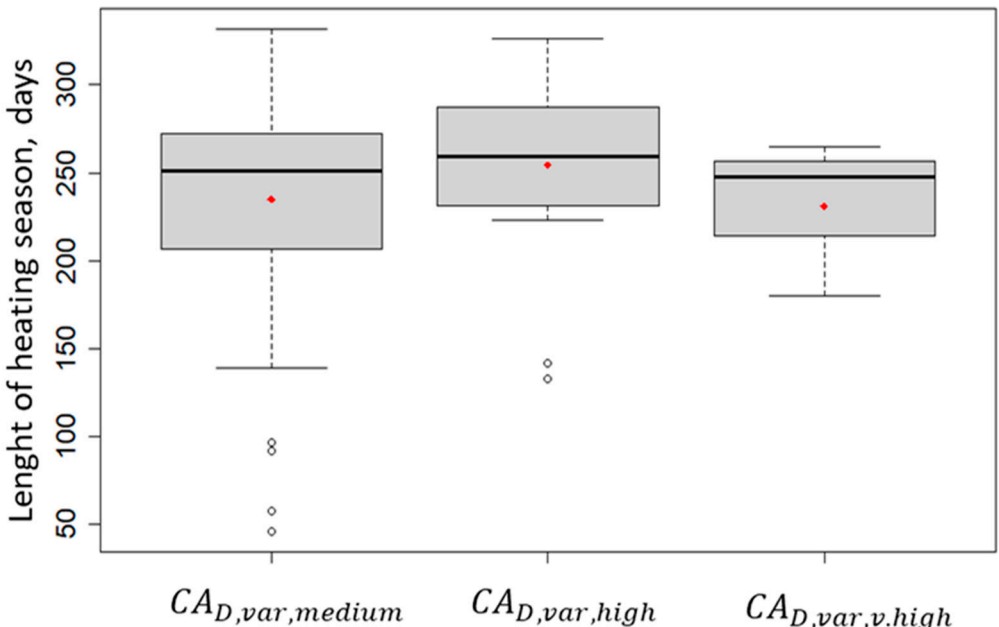

**Figure 11.** Length of the heating season from the models of the SH operation intensity.

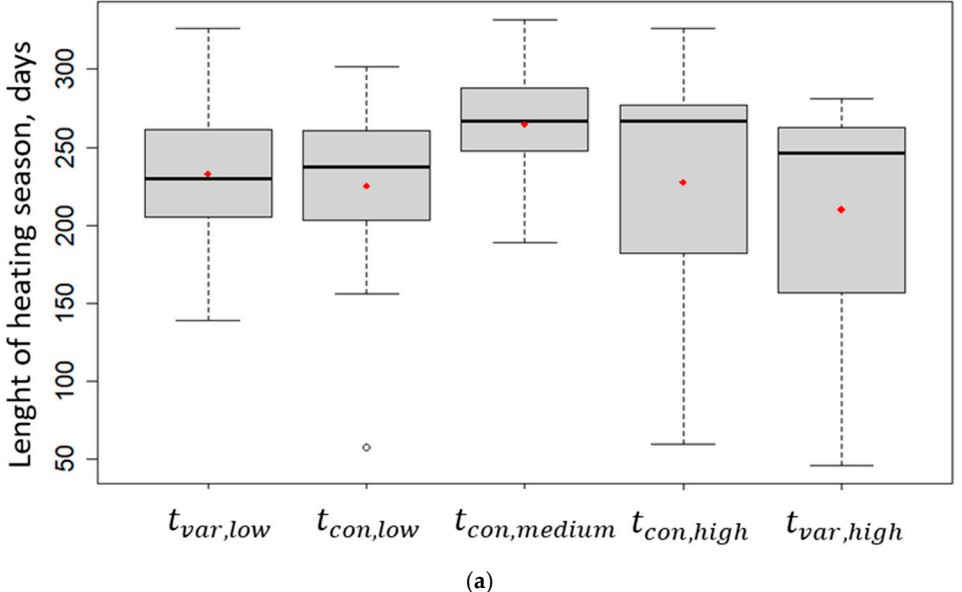

(**a**)

**Figure 12.** *Cont.*

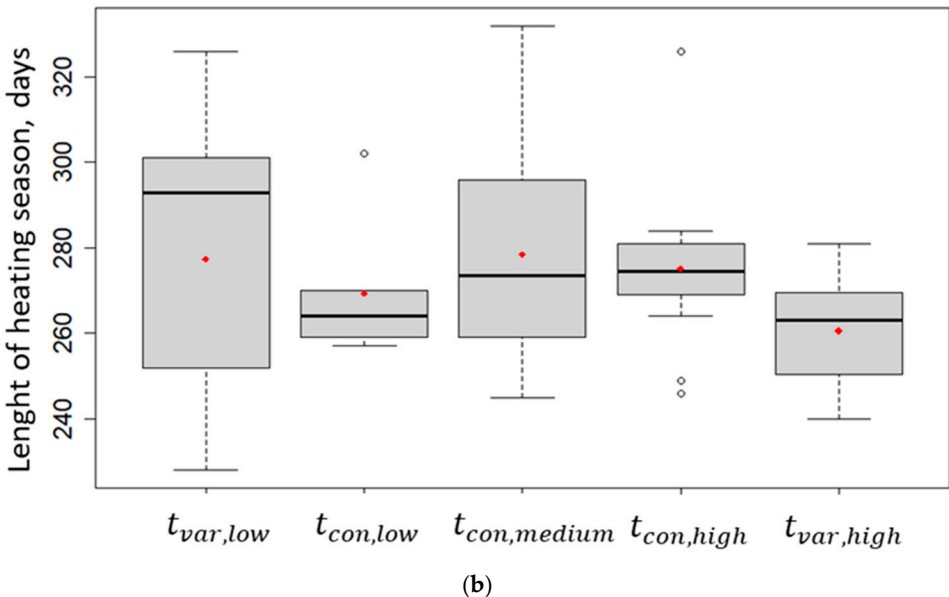

**(b)**

**Figure 12.** Length of the heating season from the indoor temperature models: (**a**) all the models and (**b**) the most reliable models (where RMSE < 25%).

## 4. Discussion

### 4.1. Sources of Uncertainty

The first issue to consider is how reliable the presented method is. As noted in the Introduction, the results provided here are preliminary. For a complete analysis of this approach, an experiment is needed in which, in addition to data from the heat meters, there will be data on the room temperature, the radiator temperature, and the window openness. It would also be valuable to have better-prepared heat meters with a higher resolution (see Section 2.3.1). It will then be possible to interpret precisely how the values obtained from this approach relate to the room conditions and the radiator operation, for example, whether $t_{in}$ in Equation (1) is correlated with the room operative temperature or whether this correlation is strongly disturbed by radiator location. This is a challenge for all such indirect analyses that relate to an energy signature [27].

Second, it is important to note the specifics of the system under study. This is a constant-temperature heat network due to using the RTS, with direct coupling between the heat source and the radiators. The year-round high value of $t_{SH,sup}$ makes the operation of the TRV, when maintaining a fixed set point position (i.e., during the automatic flow control), less stable. For a given position, the TRV has a defined characteristic relating to the dependence of the opening degree and the temperature of the integrated sensor (i.e., the control stem). The flow rate and, thus, the radiator power at a constant, $t_{SH,sup}$, is a function of the opening degree of the TRV and the pressure generated by the pump in the network. If the pump operates in constant-pressure mode (which is how it is set up in this installation), then the power of the radiator can be assumed to depend on $t_{in}$, regardless of the heating load of the room. Next, the decrease in the power occurs as a result of an increase in $t_{in}$ and only after another decrease will $t_{in}$ begin to increase. In the case of the variable $t_{SH,sup}$, as a function of the external temperature, using the heating (TRVs have a hysteresis curve that stabilizes the operation slightly: on an increase in $t_{in}$, a given value corresponds to a lower flow than the same value during a decrease of $t_{in}$ [48]) curve, the operation of the TRV would be more stable. Then, the heater power will also be regulated by the change in $t_{SH,sup}$ and, thus, correlates with the room heat load. Due to the constant $t_{SH,sup}$, a manual control enables higher comfort compared with the automatic TRV operation. Thirdly, in the case of the flow systems, the pressure system in the pipes and the proper balancing of the overall system are important [20,48]. In this aspect, the system was not evaluated.

At the end of this Section, it is noteworthy that the model that best describes the total consumption of all the metered dwellings is consistent with expectations. The following models describe it:

- $CA_{D,var,medium}$, in which the control is performed by changing the heating area or the operating time of the radiators, with maximum utilization of the available heating area, $\sum A_{D,max}$, in peak consumption. This is without increasing the heat transfer coefficient, $C$, between the radiator and the room.
- $t_{con,medium}$ representing the average internal temperature values observed in similar buildings during direct measurements [47].

### 4.2. The Actual Use of TRVs and Its Implications

As the Introduction notes, using the TRVs does not guarantee energy savings [18]. Although some datasets in the literature support this possibility [20], it is unclear whether energy savings can be achieved by balancing the system with a TRV or by automated control of the room temperature. However, a common problem with TRVs is their acceptance by users. As Kempton [49] noted over 30 years ago, the user perception and the use of thermostats can be divided in two ways. Kempton formulated them as a feedback theory and a valve theory. The first method of perception is consistent with the technical properties of the thermostats and accounts for the mechanism of their operation. The second method treats the thermostats as devices that directly serve to activate a heat (or cooling) source, perhaps with a specific power, but not self-regulating as a function of a set temperature. According to Kempton, 25–50% of Americans follow the valve theory. It can be assumed that after so much time has passed since this research, this state has changed in favor of the feedback theory as a result of public education. However, is it a matter of the education of society or more a matter of perception and the needs of individual people, for which the age of technology is less important? Occupant behaviors consistent with valve theory were also observed by Galvin [50] in thermostat data from Germany collected in 2012 and Aragon et al. in thermostat data from elderly residents collected in the UK in 2019/2020 [25]. A different picture is shown by Karjalainen's [22] research on how thermostats (including TRVs) were used in office buildings and homes in Finland, which was conducted about 15 years ago. They show that more than 60% of people in the living room adjust thermostats less than once a month and 80% less than once a week, but valve theory would suggest a more frequent adjustment of the settings. The response to feeling cold is usually to dress warmer (for more than 50% of the cases), only around 20% first increase the thermostat setting. This is more of a savings strategy, as Karjalainen reported no major problems with the technology: the Finns are highly rated, at 4.5/5 on average, for the accessibility of these thermostats and the ease of changing their settings. Only less than 20% felt that the room temperature response to the change in settings was slow or very slow.

The results presented in this paper are more supportive of Kempton's observations, with almost all the dwellings best described by the model that assumes manual control of the radiator area: increasing heat loads are covered by either running more (area) radiators or operating them for more extended periods. Flow control, which simultaneously causes lower flow rates and a higher decrease in heating water temperature as it flows through the system, is not only invisible in the surveyed buildings but also in the vast majority of dwellings. The discrepancy between the control methods declared by Karjalainen's questionnaires [22] and those resulting from the present analysis, on the assumption that both studies are reliable, can be explained by the occupancy of the dwellings. According to research [51], in Poland, the degree of overpopulation of apartments is four times higher than in Finland; in the studied housing estates, the population density was at least 22 m$^2$ per person. This situation causes the internal heat gains to have a much greater share in the energy balance of the dwelling and significantly affects the dynamics of this balance on the heat load side. This favors the dynamics in the use of the TRV and windows (natural ventilation) to obtain the effect faster without waiting for the automatic response of the TRV. A similar explanation could be valid for research by Galvin [50], which was performed on

data from low-energy houses, so internal heat gains also have a greater share in the energy balance than in the case of standard buildings. As a result, the occupants cover their needs by not using TRV controllability.

From an energy efficiency standpoint, a failure to utilize the TRV automatic control functionality of the system is a problem. Such a situation is associated with higher flows in the system, i.e., a larger pump energy consumption, alongside a high heating medium temperature that leads to higher distribution losses. Moreover, radiators reach higher temperatures that cause higher ventilation heat losses when windows are open and higher heat losses by more intensive heat transfer to internal surfaces of the building envelope. This heat does not warm the space but directly dissipates to the outdoor environment.

*4.3. Presented Approach Application*

Assuming the robustness of the presented model, it could deliver crucial information about occupant-heating system interactions. These data could act as a basis for feedback provided to occupants, which will be the first step to educating or helping them toward more efficient use of their heating system. The next steps could be in two directions. One is the use of data to share the heat consumption bill between dwellings in one building. So far, methods based on dwelling heat meters or radiator heat allocators neglect heat transfer between dwellings that could be estimated based on the presented model, as one of the model outputs is the indoor temperature. The second possibility is to use information from the model to control the dwelling heating systems more individually, but such an approach needs to be developed with a deeper understanding of the user expectations regarding the heating system.

## 5. Conclusions and Future Work

The method for analyzing the daily data from heat meters, which includes the volume of heat delivered to the apartment and the volume of heating water that flows through the residential system, is presented. Based on these data, the paper answers two research questions:

(1) What information regarding the occupant control of the SH can be extracted from heat meters?

A novel approach for heat meter data analysis is presented. The employed model for the SH system, which is fitted to gather a relevant dataset, enables us to estimate indoor temperatures maintained in the dwelling, the intensity of the use of the radiators, and the way that the TRV is used. The analysis resulted in the description of the dwellings using the SH model parameters and examined the correlation of the model parameters with the characteristics of the dwellings. Two correlations were observed. The first correlation was between the estimated internal dwelling temperature and the length of the heating period of the dwelling. The median length of the heating period was lower for dwellings in which the lower internal temperatures were predicted (Figure 12b). The second correlation relates to the estimation by the model of the internal dwelling temperature and the building name to which particular apartments belong (Table 3). Observed higher estimated internal temperatures in building WE3 could be caused by large heat gains from the boiler room located in that building, which disturbed the SH usage in some of the dwellings of this building.

(2) Is it possible, based on this information, to learn more about people's use of TRV?

The model of the SH system used was fitted to gather a dataset that enables an estimation of the intensity of the use of the radiators and the way that the TRV is operated. The research indicated that the data from the vast majority of the dwellings (i.e., 96%) do not match the model of the automated control TRV, which means that users employed the TRV according to valve theory [49], i.e., without account for the important functionality of these devices, which self-adjusts as a function of the set indoor temperature.

The additional output of the presented research, given in Supplementary Material S1, is a method for separation of the registered heat consumption on the SH and the DHW consumption, accounting for the water system supply temperature during the year as well

as the variation of the DHW consumption during the week (i.e., weekdays, Saturdays, and Sundays).

The presented method needs further improvement by a more comprehensive research project. There are two possible paths. The first is to employ laboratory space [37] or one-room measurements and observe the heat and water flow of a single radiator. In such experiments, the ventilation air temperature and its flow could be modified as radiator surroundings (e.g., furniture or location of ventilation air inlet) as well as water pressure and temperature in the heating system. The results could be used to compare the estimated $CA_D$ and $t_{in}$ to real values or ventilation and surround modifications. The second approach is to conduct the presented analysis in a building where smart TRV is implemented, e.g., ref. [25], and verify if the recorded use of the TRV could be found in heat meter data for the whole dwelling/building.

The improved model will be an easy-to-use tool to analyze the use of SH, which can be employed to control (or tariff) these systems or to advise users. The results of the study on the use of the TRV suggest a need to rethink the radiator control for multi-family buildings in thermally renovated and newly constructed buildings.

**Supplementary Materials:** The following supporting information can be downloaded at: https://www.mdpi.com/article/10.3390/en16227485/s1, Supplementary Material S1: estimation of water supply system temperature; estimation of supply temperature for residential thermal station and space heating; algorithm for separation of heat consumption for space heating and domestic hot water. Supplementary Material S2: the graphical comparison of the model results with the measurements for each dwelling.

**Author Contributions:** Conceptualization, K.B.; methodology, K.B.; investigation, K.B.; resources, K.B.; data curation, K.B.; writing—original draft preparation, K.B.; writing—review and editing, K.B. and A.G. and H.K.; visualization, K.B.; supervision, A.G. and H.K. All authors have read and agreed to the published version of the manuscript.

**Funding:** This research received no external funding. The APC was funded by Poznań University of Technology grant number SBAD PB 0981 and SBAD MK 0980.

**Institutional Review Board Statement:** Not applicable.

**Informed Consent Statement:** Not applicable.

**Data Availability Statement:** Part of data (heat meters data) are available from the ASHRAE Global Occupant Behavior Database: https://ashraeobdatabase.com (accessed on 26 October 2023).

**Acknowledgments:** The authors would like to thank our students, the manager of the investigated building, and the service staff for supporting this research. Special thanks also to Veolia Energia Poznań SA for data collected from their heating network and Kamstrup Sp. z o.o. for support in data collection.

**Conflicts of Interest:** The authors declare no conflict of interest.

## Nomenclature

**Abbreviations**

| | |
|---|---|
| DHW | domestic hot water |
| HN | heat network |
| RTS | residential thermal station |
| SH | space heating |
| TRV | thermostatic radiator valve |
| WSS | water system supply |

**Dates**

| | |
|---|---|
| $d_{V,0}$ | start of general vacation |
| $d_{V,1}, d_{V,2}$ | start and end date of school vacations |
| $d_{SH,0}, d_{SH,1}$ | estimated start and end of the heating season |

**Temperatures**

| | |
|---|---|
| $t_{DHW}$ | DHW temperature at the tapping point, °C |
| $t_{in}$ | daily mean indoor temperature of the apartment, °C |
| $t_{HN,ret}$ | daily mean return temperature of HN, °C |
| $t_{HN,sup}$ | daily mean supply temperature of HN, °C |
| $t_{out}$ | daily mean outdoor temperature, °C |
| $t_{RTS,sup}$ | daily mean supply temperature of RTS, °C |
| $t_{SH,sup}$ | daily mean supply temperature of SH, °C |
| $\Delta t_{SH-in,log}$ | daily mean logarithmic temperature difference between the radiator temperature and the indoor temperature of the dwelling, °C |
| $t_{WSS}$ | daily mean WSS temperature on RTS supply, °C |

**Energy, Power, and Flow**

| | |
|---|---|
| $\dot{m}_{D,\,SH}$ | mean daily mass flow of heating water in SH of the dwelling, kg/s |
| $Q_D$ | daily heat consumption of the dwelling, J |
| $Q_{D,DHW,wd}$, $Q_{D,DHW,st}$, $Q_{D,DHW,sn}$ | daily heat consumption of the dwelling for DHW on weekdays, Saturdays, and Sundays, respectively, J |
| $Q_{D,SH}$ | daily heat consumption of the dwelling for SH, J |
| $\dot{Q}_D$ | mean daily heating power of RTS and radiators in the dwelling, W |
| $\dot{Q}_{D,SH}$ | mean daily heating power of SH (sum of radiators power) in the dwelling, W |
| $\dot{Q}_{D,SH,max}$ | maximum heating power of SH (sum of radiators power) in the dwelling, W |
| $\dot{Q}_{D,SH,meas}$ | measured mean daily heating power of SH (sum of radiators power) in the dwelling, W |
| $\dot{Q}_{HN}$ | mean daily heating power of HN, W |
| $\dot{Q}_{HN,loss,sup}$ | mean daily heat loss from the supply pipe of HN, W |
| $V_D$ | daily heating water flow through the dwelling, m³ |
| $V_{D,DHW,wd}$, $V_{D,DHW,st}$, $V_{D,DHW,sn}$ | daily heating water flow in dwelling RTS for DHW, on weekdays, Saturdays, and Sundays, respectively, m³ |
| $V_{D,SH}$ | daily heating water flow in the dwelling for SH, m³ |
| $\dot{V}_{HN}$ | mean daily volume flow of heating water in HN, m³/s |

**Other Physical Variables**

| | |
|---|---|
| $A_D$ | heating area of the radiator, m² |
| $A_{D,max}$ | total heating area of the radiators in the dwelling, m² |
| $A_f$ | floor area of dwelling, m² |
| $A_{win}$ | window area, m² |
| $C$ | coefficient of the intensity of heat exchange between the radiator and the surrounding environment, W/(m²K) |
| $C_m$ | thermal capacity of dwelling walls, kJ/K |
| $c_w$ | specific heat of water, J/(kgK) |
| $H_{tr}$ | heat transfer coefficient of the dwelling envelope, W/K |
| $k_{Htr}$ | correction factor $\dot{Q}_D$, which accounts for dwellings and staircases without heat meters, unitless |
| $k_{q(t_{WSS})}$ | coefficient concerning the influence of $t_{WSS}$ on the change in heat consumption for DHW, unitless |
| $k_{v(t_{WSS},\,t_{RTS,sup})}$ | coefficient concerning the influence of $t_{WSS}$ and $t_{RTS,sup}$ on the change in heating water flow for DHW, unitless |
| $n$ | exponent of the thermal characteristics of the radiator, unitless |
| $\rho_w$ | water density, kg/m³ |

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
