# Peer review of "Radiators Adjustment in Multi-Family Residential Buildings—An Analysis Based on Data from Heat Meters"

_energies, doi:10.3390/en16227485_

Round 1

Reviewer 1 Report

Comments and Suggestions for Authors

The scientific paper exhibits potential; however, it requires in my opinion significant revision, particularly in the literature review and conclusion sections. The paper lacks clarity in presenting a comprehensive review of existing literature, establishing a clear literature gap, with all the necessary references, and establishing a strong connection between the literature review and the research question. Additionally, the conclusion should be improved by explicitly restating the research question and providing a more direct discussion of how the findings address it.

Specific Points:

  1. Citations in Lines 88-89 and 91-92: Lines 88-89 and 91-92 require additional references. The paper mentions concepts related to comfort and certain claims, but it's essential to provide citations to substantiate these assertions. A more comprehensive literature review in these areas is necessary to establish the context and relevance of the research.
  2. Specification of Scopus Research in Line 95: Line 95 mentions "Scopus research" without specifying the date. It is crucial to provide the publication date of the Scopus research to inform the readers about the timeliness and relevance of the data used.
  3. Citations in Lines 98-101: Lines 98-101 contain statements that lack references. It's unclear whether these are hypotheses to be addressed in the paper or if there is existing literature to support these points. .
  4. Acronym Clarifications: In Line 114, the acronym "SH" should be extended the first time it appears, and the same should be done for "TRV" in Line 120. This ensures that readers are not confused by unfamiliar acronyms.
  5. Introduction and Acronyms: The introduction should provide a more concise and clear definition of the field and the research topic. Additionally, the excessive use of acronyms throughout the paper, without proper definitions, makes it challenging to follow. Consider providing definitions when acronyms are introduced.
  6. Literature Review Depth and Clarity: The literature review needs to be significantly revised for clarity and depth. Ensure that it clearly identifies the gaps in the existing literature and establishes a strong connection between the literature review and the research question. Pay attention to English fluency and ensure that proper references are cited.
  7. Consideration of Measurement Dates: Since the measurements used in the paper are from 2015 and 2016, it's essential to discuss any potential implications of using older data. Address how these dated measurements may affect the relevance and validity of your findings.
  8. Clarification of Lines 206-210: Lines 206-210 lack clarity. The paper should explicitly state the amount of data used in the research and elaborate on how the identified limitations affected the analysis and data utilization.
  9. Improvement in the Conclusion Section: In the conclusion section, it is advisable to recall the research question and engage in a more direct discussion of how the findings contribute to addressing it. Strengthen the connection between the research question and the obtained results.

In conclusion, the paper requires major revisions, particularly in the literature review section and the conclusion. Enhancing clarity, depth, and the connection between the literature and the research question will significantly improve the overall quality of the paper. Proper citations, clearer explanations, and well-defined acronyms are key areas to focus on for improvement.

Comments on the Quality of English Language

The paper could benefit from language refinement, particularly in its more discursive sections (Introduction, literature review and discussion). This could enhance readability and engagement with the academic audience. In particular the fluency in the literature review is not enough clear and do not clearly conduct the reader from the identification of gaps and then of research questions to be addressed.

Author Response

Dear Reviewer,

Thank you very much for your time and effort to help us better communicate our achievement. It is encouraging that you see potential in our research. Based on your remarks we correct some part of the paper, complete introduction and rearrange conclusion. Regard language comments we correct some wrongly written sentences, but basically the text have been corrected by native speaker with PhD therefore we consider it on appropriate level. If you notice any not clear parts, pleas point them out. We do our best to correct them. The detail are described below and presented (by change tracking mode) in new version of paper.

  1. Citations in Lines 88-89 and 91-92: Lines 88-89 and 91-92 require additional references. The paper mentions concepts related to comfort and certain claims, but it's essential to provide citations to substantiate these assertions. A more comprehensive literature review in these areas is necessary to establish the context and relevance of the research.

A: Thank you for this point. Yes, we mention about importance of human-building interaction but we didn’t refer to papers where result of neglecting this issue are described. Therefore we rephrased text and referred two papers which papers which presents result of neglecting human-building interaction.

  1. Specification of Scopus Research in Line 95: Line 95 mentions "Scopus research" without specifying the date. It is crucial to provide the publication date of the Scopus research to inform the readers about the timeliness and relevance of the data used.

A: Indeed, the date of search is crucial parameter. We have conducted search one more time, update paper number and add data.

  1. Citations in Lines 98-101: Lines 98-101 contain statements that lack references. It's unclear whether these are hypotheses to be addressed in the paper or if there is existing literature to support these points.

A: We one more time read this paragraph and noticed that your point. Thank for this! We decide to delate this part of introduction as it is digression discussion which not help to present motivation to for taking up the topic.

  1. Acronym Clarifications: In Line 114, the acronym "SH" should be extended the first time it appears, and the same should be done for "TRV" in Line 120. This ensures that readers are not confused by unfamiliar acronyms.

A: The paper finally use a lot of symbols and acronyms, therefore our explanation strategy should be efficient. Thank you for pointing this out. We unify the way in which acronyms are introduced and used. Now their definition is next to them where they first time appear in the abstract, nomenclature, the main text or Annex 1.

  1. Introduction and Acronyms: The introduction should provide a more concise and clear definition of the field and the research topic. Additionally, the excessive use of acronyms throughout the paper, without proper definitions, makes it challenging to follow. Consider providing definitions when acronyms are introduced.

A: As we mention above we unify acronym explanation approach. Also introduction have been modified – we hope: in good direction.

  1. Literature Review Depth and Clarity: The literature review needs to be significantly revised for clarity and depth. Ensure that it clearly identifies the gaps in the existing literature and establishes a strong connection between the literature review and the research question. Pay attention to English fluency and ensure that proper references are cited.

A: Than you for this comment. We try to modify introduction in according to this guidelines. Now, we hope, research questions are clearly stated and supported by literature review.

  1. Consideration of Measurement Dates: Since the measurements used in the paper are from 2015 and 2016, it's essential to discuss any potential implications of using older data. Address how these dated measurements may affect the relevance and validity of your findings.

A: We regret that the data waited so long to be published. Your idea to prove that it is not “overdue” is helpful – thank you. We decide to show it by additional figure (Fig. 5) where outdoor temperature for neighbor years are compared. We also comment it in section 2.3.2. Outdoor air temperature.

  1. Clarification of Lines 206-210: Lines 206-210 lack clarity. The paper should explicitly state the amount of data used in the research and elaborate on how the identified limitations affected the analysis and data utilization.

A: Thank you for this point. We think that this was just presented in paper, but we have added explicitly reference to relevant figure, where low dwelling heat consumption is correlated with increase of model low accuracy (see section 2.3.1. Heat consumption and heating water flow).

  1. Improvement in the Conclusion Section: In the conclusion section, it is advisable to recall the research question and engage in a more direct discussion of how the findings contribute to addressing it. Strengthen the connection between the research question and the obtained results.

A: Thank you for this advice. We rephrased conclusion section acc. to answers to research questions.

Kind regards

Reviewer 2 Report

Comments and Suggestions for Authors

The authors' research offers a fresh perspective on understanding how residents interact with space heating systems and explores the potential of utilizing calorimeter data for analysis. After reviewing this paper, I have the following comments and suggestions, hoping they will assist you in further improving the manuscript.

1. The use of abbreviations in the text should be standardized. For instance, "Af" in Table 1 is not defined in the Nomenclature section. Also, the abbreviation "SH" has been mentioned several times, yet on line 186, author have again introduced "space heating system (SH)."

2. The finding that most users tend not to utilize the automated adjustment features and prefer manually operating Thermostatic Radiator Valves (TRVs) is significant. A more in-depth discussion on this behavioral pattern would be beneficial, such as, what drives this preference? How does this affect SH efficiency and comfort levels of the inhabitants? Moreover, it would be constructive to discuss in detail how the improved model you propose can be implemented in practical applications and how these implementations can enhance energy efficiency and comfort for users.

3. The need authors mentioned for further refinements in research methods indicates there's more work to be done. In this section of the paper, more specifics on how the anticipated research would impact the current model, including the anticipated experimental design, data collection methods, and how to address the technical specificities not touched upon in the current research would be beneficial.

Author Response

Dear Reviewer,

Thank you very much for your time and effort to help us better communicate our achievement. It is encouraging that you see our approach as fresh perspective. Based on your remarks we correct some part of the paper and complete discussion. If you notice any not clear parts, pleas point them out. We do our best to correct them. The detail are described below and presented (by change tracking mode) in new version of paper.

  1. The use of abbreviations in the text should be standardized. For instance, "Af" in Table 1 is not defined in the Nomenclature section. Also, the abbreviation "SH" has been mentioned several times, yet on line 186, author have again introduced "space heating system (SH)."

A: The paper finally use a lot of symbols and acronyms, therefore our explanation strategy should be efficient. Thank you for pointing this out. We unify the way in which acronyms are introduced and used. Now their definition is next to them where they first time appear in the abstract, nomenclature, the main text or Annex 1. Except “Af” we have found one more abbreviation in Table 1 which was not consistent with Nomenclature. It is corrected.

  1. The finding that most users tend not to utilize the automated adjustment features and prefer manually operating Thermostatic Radiator Valves (TRVs) is significant. A more in-depth discussion on this behavioral pattern would be beneficial, such as, what drives this preference? How does this affect SH efficiency and comfort levels of the inhabitants? Moreover, it would be constructive to discuss in detail how the improved model you propose can be implemented in practical applications and how these implementations can enhance energy efficiency and comfort for users.

A: Thank you for this comment. We think that we answer on this questions in the two last paragraphs of section 4.2. The actual use of TRVs and its implications and in new section 4.3. Presented approach application

  1. The need authors mentioned for further refinements in research methods indicates there's more work to be done. In this section of the paper, more specifics on how the anticipated research would impact the current model, including the anticipated experimental design, data collection methods, and how to address the technical specificities not touched upon in the current research would be beneficial.

A: Than you for this advice. We hope that new text of section 5. Conclusion and future work, especially penultimate paragraph improve our idea description.

Kind regards